# The Role of Genetic, Metabolic, Inflammatory, and Immunologic Mediators in the Progression of Intraductal Papillary Mucinous Neoplasms to Pancreatic Adenocarcinoma

**DOI:** 10.3390/cancers15061722

**Published:** 2023-03-11

**Authors:** Kylie E. Shockley, Briana To, Wei Chen, Gerard Lozanski, Zobeida Cruz-Monserrate, Somashekar G. Krishna

**Affiliations:** 1Department of Internal Medicine, The Ohio State University Wexner Medical Center, Columbus, OH 43210, USA; 2Department of Pathology, The Ohio State University Wexner Medical Center, Columbus, OH 43210, USA; 3Division of Gastroenterology, Hepatology, and Nutrition, and The James Comprehensive Cancer Center, The Ohio State University Wexner Medical Center, Columbus, OH 43210, USA

**Keywords:** PDAC, IPMN, pancreatic cyst, genetic alterations, metabolic alterations, KRAS, GNAS

## Abstract

**Simple Summary:**

Intraductal papillary mucinous neoplasms (IPMN) are benign pancreatic cysts found in the ducts of the pancreas that have the potential to become malignant. Identifying IPMNs that have high potential to become pancreatic cancer may help prevent unnecessary surgery which is the definitive treatment of IPMNs. Currently, the management of IPMNs are dependent on variable factors including characteristics of the cyst, size of the pancreatic duct, and the presence or absence of obstructive jaundice. Identifying potential biomarkers may help more accurately distinguish high risk IPMNs from low risk IPMNs that do not need surgical intervention. This review summarized the various changes within IPMNs that promotes their progression to pancreatic adenocarcinoma. In addition, this review highlighted potential biomarkers that can distinguish IPMNs that have a high risk of becoming cancerous.

**Abstract:**

Intraductal papillary mucinous neoplasms (IPMN) have the potential to progress to pancreatic ductal adenocarcinoma (PDAC). As with any progression to malignancy, there are a variety of genetic and metabolic changes, as well as other disruptions to the cellular microenvironment including immune alterations and inflammation, that can contribute to tumorigenesis. Previous studies further characterized these alterations, revealing changes in lipid and glucose metabolism, and signaling pathways that mediate the progression of IPMN to PDAC. With the increased diagnosis of IPMNs and pancreatic cysts on imaging, the opportunity to attenuate risk with the removal of high-risk lesions is possible with the understanding of what factors accelerate malignant progression and how they can be clinically utilized to determine the level of dysplasia and stratify the risk of progression. Here, we reviewed the genetic, metabolic, inflammatory, and immunologic pathways regulating the progression of IPMN to PDAC.

## 1. Introduction

Pancreatic ductal adenocarcinoma (PDAC), currently the third leading cause of cancer-related mortality, is projected to be the second most common cause of death due to malignancies in the United States by 2030 [1]. This poor clinical outcome can be partly attributed to the often vague or absent clinical symptoms associated with pancreatic ductal adenocarcinoma. Thus, many patients with PDAC present with advanced stages of the disease. While early PDAC, which constitutes <15% of all new diagnoses, are surgically resectable and associated with a better outcome, the mainstay treatment of advanced pancreatic cancer is limited to chemotherapy. Despite extensive research, attempts to develop targeted therapies were largely unsuccessful [2]. A better understanding of the development of PDAC may provide insight into the discovery of new diagnostic tools and therapeutics. PDAC commonly arises from precursor lesions including pancreatic intraepithelial neoplasia (PanIN), intraductal papillary mucinous neoplasm (IPMN), and mucinous cystic neoplasm (MCN). Among the cystic precursors, IPMN is the most prevalent.

These cysts are mucin-producing and arise from the main pancreatic or the branch duct. Main duct lesions (MD-IPMN) has a higher risk of malignancy compared to branch duct (BD-IPMN). There is a third type that shares features of MD-IPMN and BD-IPMN and is referred to as mixed type (MT-IPMN). MT-IPMN and MD-IPMN tend to be more symptomatic. Symptomatic patients tend to progress to malignancy more than those who are asymptomatic, regardless of subtype. While they all arise from cells which produce mucin, four histopathological IPMN types are distinguished by the specific mucin(s) they produce: gastric (49–63%), intestinal (18–36%), pacreaticobiliary (7–18%), and oncocytic (1–8%). Of these, gastric is the most common and rarely progresses to malignancy. While pancreaticobiliary is less common, it is commonly associated with aggressive PDAC [3]. The degree of cytological atypia and crowding of the epithelium allows for classification of the IPMNs into low, intermediate, and high grade dysplasia [4]. In addition to MD-IPMN carrying a larger risk for malignant progression, other high risk features inform malignant potential including the size of the lesion, rate of growth of the lesion, presence of solid components within the lesion, the presence of high grade dysplasia, main duct dilation on imaging, and the presence of symptoms including jaundice, new onset diabetes, or pancreatitis [3].

The primary management of IPMN is the surgical resection of high-risk lesions that can progress to malignant disease [5]. Risk stratification of these cystic lesions is largely based on imaging and clinical characteristics of the cyst including size, location, and grade of dysplasia [5]. The exact molecular mechanisms driving IPMN progression to PDAC are unclear; however, genetic, metabolic, immune, and inflammatory changes appear to play a role in this process (Figure 1). The following review summarizes the genetic drivers and metabolic events that potentially contribute to IPMN tumorigenesis and progression to malignancy. In addition, the role of inflammatory and immune modulation in the development and progression of IPMN are highlighted. Finally, the review concludes with a look at the clinical applications that can help provide prognostic insights and potentially guide the management of IPMNs. 

## 2. Genetic Alterations

Whole exome and targeted sequencing were utilized to better characterize the genetic alterations in IPMNs. The most commonly mutated gene in IPMNs is *KRAS* with 50–80% of IPMNs harboring a *KRAS* mutation [6,7]. The most common location of the *KRAS* mutation is in codon 12, resulting in a G12D mutation. This *KRAS* mutation encodes a constitutively active GTP-binding protein that regulates various signaling cascades including cell growth and proliferation. Furthermore, *KRAS* mutations are found in >90% of PDACs and, hence, are thought to be essential and an early event for tumorigenesis [8]. Mutation in the *GNAS* gene is another common alteration in IPMNs and was identified in 40–70% of lesions [9]. *GNAS* encodes the stimulatory alpha subunit of the guanine nucleotide-binding protein which activates cyclic adenosine monophosphate (cAMP), leading to the activation of multiple effectors including protein kinase A and EPAC (Exchange Protein directly activated by cAMP). Although the relevance of these pathways in pancreatic cancer was not fully established, these pathways were implicated in other cancer types [10,11]. The *GNAS* mutation is predominantly located at codon 201, commonly leading to R201H or R201C alterations [9]. This is thought to be an activating mutation that results in the constitutive activation of *GNAS*. In contrast to *KRAS*, *GNAS* mutations are not commonly observed in non-IPMN PDACs but were found in 25–61% of IPMN- derived PDACs [7,12,13].

*KRAS* and/or *GNAS* mutations are found in >90% of IPMNs, both in those with advanced neoplasia (high-grade dysplasia or adenocarcinoma) and low-grade dysplasia [7]. The prevalence of their alteration suggests that they play an important role in IPMN development. These mutations were found to coexist, suggesting that they are not mutually exclusive. Interestingly, transgenic mouse models of IPMNs required synergistic mutation between *GNAS* and *KRAS* to generate cystic lesions that are histologically similar to human IPMNs [9,14]. Although there are no significant differences in *KRAS* and *GNAS* alterations in IPMNs with and without advanced neoplasia, a difference in mutation frequency were observed when analyzing the histological subtypes [7].

IPMNs can be divided into histological subtypes including gastric, intestinal, and pancreatobiliary. The oncocytic subtype is now considered a separate entity, i.e., “Intraductal Oncocytic Papillary Neoplasm”, by WHO’s classification of tumors, due to its different molecular and clinical features. These subtypes can give rise to different types of invasive carcinoma [15]. In general, intestinal IPMN gives rise to colloid carcinoma while tubular carcinoma can arise from gastric or pancreatobiliary subtype. Interestingly, there is a higher frequency of *GNAS* mutation in IPMN-associated colloid carcinoma and *KRAS* mutations in IPMN-associated tubular carcinoma [7]. This difference in mutational frequency may be clinically relevant, given that tubular adenocarcinoma is associated with worst clinical outcomes, and thus, it may offer prognostic value.

Inactivating mutation in the *RNF43* gene is another common genetic alteration found in IPMNs, occurring in 10–75% of cystic lesions [7]. RNF43 encodes a transmembrane E3 ubiquitin ligase that negatively regulates the Wnt pathway. Thus, the loss of function of RNF43 confers Wnt pathway activation, which was shown to play a potential role in mediating PDAC progression [16,17]. Other loss of function mutations found in IPMNs include *TP53, CDKN2A*, and *SMAD4*. These genetic alterations occur at a lower rate in IPMNs with low-grade dysplasia compared to those with advanced neoplasia, suggesting that these genes may play a role in mediating progression to malignancy [7]. Furthermore, these genetic alterations are also commonly detected in invasive pancreatic cancer.

More recently, Noe et al. performed whole exome sequencing (WES) of IPMNs and MCNs and their associated invasive carcinomas to better understand the genetic alteration driving tumorigenesis in these cystic precursors [18]. WES analysis revealed the high prevalence of previously identified drivers of pancreatic tumorigenesis including *KRAS*, *GNAS*, *RNF43*, *CDKN2A*, *TP53*, and *SMAD4*. In addition, the use of WES revealed novel mutations in IPMN tumorigenesis. Somatic mutations in the *ATM* gene were found in 17% of lesions. *ATM* encodes a serine/threonine kinase that is involved in DNA double-strand break repair as well as other cellular processes including metabolism and chromatin remodeling. Moreover, individuals with germline pathogenic *ATM* variants appear to have an increased lifetime risk of pancreatic cancer [19,20]. In addition, loss of ATM expression in a mouse model of PDAC resulted in a greater number of proliferative precursor lesions [21]. Somatic mutation in *GLI3* gene was also observed in 8% of samples. *GLI3* gene encodes a transcription factor that is a member of the Hedgehog signaling pathway and is involved in normal cellular processes including tissue and immune cell development. Furthermore, studies implicated GLI3 in mediating pro-tumorigenesis pathways in various cancers, including pancreatic cancer [22,23,24].

## 3. Aberrant Methylation

In addition to mutational changes, gene expression can be altered through DNA methylation. This form of epigenetic modification involves adding a methyl group to the fifth carbons of the cytosine ring of DNA. DNA is primarily methylated at CpG islands which are GC-enriched regions. When DNA methylation occurs on a gene promoter, it can inhibit transcriptional regulators from binding and initiating transcription. Alternatively, DNA methylation can lead to an alteration in chromatin structures, typically resulting in a repressive state. Methylation profiling of IPMNs revealed aberrant methylation of at least one CpG island in >80% of IPMNs [25]. Furthermore, the average number of methylated loci was found to be significantly increased in high-grade IPMNs vs. low grade IPMNs. CpG island methylation profiles between IPMNs and normal primary pancreatic duct samples revealed that IPMN undergoes extensive CpG hypermethylation [26]. Moreover, many of the aberrantly methylated genes are also aberrantly methylated in pancreatic cancer. Aberrant CpG hypermethylation was observed in low grade IPMNs; however, there were increasing levels of methylation with increasing levels of dysplasia. This differential hypermethylation profile based on grade of dysplasia suggest that methylation may play a role in mediating malignant progression of IPMNs. In a more recent study, Hong et al. utilized a panel of differentially methylated gene in IPMNs to predict histologic grade of cystic lesions. The methylation level of seven genes, SOX17, BNIP3, FOXE1, PTCHD2, SLIT2, EYA4, and SFRP1, in pancreatic cyst fluid samples were analyzed using methylation-specific droplet-digital PCR (dd-QMSP). All the genes, except BNIP3, were able to accurately stratify high grade vs. low grade cystic lesions with accuracies of 79.8% to 83.6% [27].

## 4. Metabolic Alterations

Genotypic changes can allow for measurable differences within the cells of IPMN. Techniques including nuclear magnetic resonance, liquid chromatography mass spectrometry, and gas chromatography mass spectrometry are all utilized to analyze what metabolites are present in cells, tissues, and, even more broadly, in blood or other body fluids. This is the basis of several biomarkers that are increased due to reprogrammed cellular metabolomics compared to an unaltered cells. For example, one of the most well-known biomarkers of pancreatic cancer, glycoprotein CA19-9, is thought to be increased secondary to perverse glycosylation and was used as a biomarker for pancreatic cancer as well as various other digestive tract cancers [28]. CA19-9 can be elevated in cases of IPMNs with pancreatic cancer; however, less frequently in those with high-grade dysplasia. So, while elevated CA19-9 is included as a worrisome feature in clinical guidelines for the evaluation of IPMN, it may not be elevated until invasive cancer is present. Elevation of CA19-9 alone cannot predict the level of dysplasia or the likelihood of progression to PDAC [29].

As an IPMN potentially progresses through a presumed stepwise pathway of low-grade dysplasia to advanced neoplasia, potential metabolites can be measured to determine the grade of dysplasia [30]. Ideally, a biomarker from a metabolic alteration would be present in the serum and be easily obtained from a patient with IPMN. The proposed biomarkers discussed in this paper do not all meet this convenience and many would have to be analyzed from IPMN cyst fluid or require further analysis of the IPMN.

In malignancy, metabolic reprogramming occurs to support increased proliferation and to support survival of altered cells. The Warburg effect, an early proposed example of this, is rapid glycolysis in response to resulting hypoxia from high energy demand not met by aerobic respiration alone [30]. Increased glucose allows for increased energy production via glycolysis in addition to increased glucose utilization in the pentose phosphate pathway, hexosamine pathway, and glycogenesis, which were all reported to be reprogrammed in cancer cells [30]. It is reasonable to think that these changes may occur prior to malignancy in cases of dysplasia. 

Histologically, glucose transporter-1 expression (GLUT-1) was detected at higher levels in poorly differentiated PDACs compared to regular pancreatic tissue. When GLUT-1 expression in IPMNs was analyzed, it was not found in low-grade dysplasia, but detected in 60% of the IPMNs with high-grade dysplasia, suggesting increased transporter expression occurs late in dysplasia prior to invasive adenocarcinoma [31]. Immunohistochemical staining of IPMN nodules, a high-risk feature, and an indication for surgical resection, also revealed significantly increased expression of SLC2A1/GLUT1 glucose transporter in samples consistent with advanced neoplasia when compared to low-grade dysplasia. There was, however, no difference between the level of expression between high-grade dysplasia and invasive cancer, suggesting that the need for increased glucose mostly occurs in the change from low- to high-grade dysplasia in the progression of IPMN to PDAC [32].

Supporting the theory that increased glucose needs are linked to increased cell growth, phosphorylated S6 ribosomal protein (pS6) was also studied with relation to GLUT1 expression. The presence of pS6 indicates mechanistic Target or Rapamycin (mTORC1) activity, which is a key coordinator for cell growth and metabolism [33]. mTOR itself is often upregulated after molecular alterations in various cancer progressions. A relationship between increased GLUT1 expression and the presence of pS6 was elucidated in pancreatic cancer cell lines. When GLUT1 expression was not elevated, levels of pS6 were similarly low and vice versa within the five different pancreatic cell lines studied. When mTORC1 and S6 phosphorylation were inhibited during this study, glucose uptake decreased, supporting the idea that one of mTORC1′s effects is increasing glucose. *KRAS* and *GNAS* are known to be mutated in IPMN tissue and such alterations were repeatedly connected to downstream activation of mTOR signaling, which, in turn, increases glucose uptake [34]. Of the known oncogenic mutations, KRAS was implicated specifically in altered metabolism in PDAC via not only regulating anabolic glucose metabolism, but reprogramming glutamine (Gln) metabolism to fuel the TCA cycle via a novel pathway [35]. Deprivation of Gln in PDAC greatly increased redox stress, and glucose and Gln were both found to be necessary for PDAC growth. KRAS allows for maintenance of the redox state of PDAC in addition to supporting the cancerous growth via altered glutamine metabolism [36,37]. In addition, KRAS signaling was also reported to remove defective mitochondria to also limit reactive oxygen species (ROS) and promote tumor growth [38]. Increased lipid scavenging was also an implicated result of KRAS mutation in PDAC, especially when fatty acid synthesis is difficult due to environmental factors of the cancer [39,40]. As KRAS was implicated as a mutation in IPMN progression to PDAC, it is reasonable to suggest downstream results from KRAS mutation in PDAC may apply to its precursor, IPMN.

Since glucose is needed for the proliferation of cancer cells, in parallel, there is increased utilization of lipids for neoplastic growth. Lipids can be utilized for energy production in glycolysis via fatty acid oxidation, or beta-oxidation, to produce ATP and NADPH, both for energy storage. In addition, lipids are utilized for the production of cellular membranes, including special glycolipids responsible for recognition and signaling molecules. The reprogramming of lipid metabolism in tumorigenesis not only helps the rapid proliferation of the cell by supporting energy needs and supplying necessary building blocks, but also can alter communication with nearby cells, affect immune surveillance, and encourage inflammation of the tumor microenvironment. Altogether, changes in lipid metabolism support the changing cellular landscape to promote neoplastic growth and proliferation [41].

When cyst fluid of resected IPMNs were analyzed, phospholipid biosynthesis, beta oxidation of very long chain fatty acids, fatty acid metabolism, oxidation of branched chain fatty acids, and other lipid pathways appeared to dwarf the activity of other metabolic pathways when compared to serous cystic neoplasms (SCN) [42]. SCNs are benign tumors with little to no malignant potential, so, in theory, would not need the increased lipid production to support dysplastic and cancerous growth. Profiles of the lipid pathways were similar between high grade dysplasia and invasive cancer. Integrated metabolite and lipidomic data from this study suggested the possibility of being able to classify the level of dysplasia, at least low- from high-grade and cancerous lesions. However, it did not perform well when high grade and malignant samples were considered separately. In addition to the increased lipidomic pathway activity, triacylglycerol (TAG) classes were altered significantly between the high-grade and low-grade dysplasia groups in both plasma and cyst fluid. When considering the cancerous group’s TAG classes and their alterations, it was most similar to the high-grade dysplasia with cyst fluid analysis, but oddly more similar to low-grade dysplasia in plasma, informing potential utilization of these TAGs as biomarkers. This study demonstrated the utility of integrating metabolite and lipidomic data for potential prediction of the level of dysplasia in IPMNs [42]. 

Altered lipoprotein processing was also observed in regard to apolipoprotein A2 (apoA2) and deviant C terminal processing associated with IPMN progression to malignancy. ApoA2 is often part of the atherosclerosis conversation, as it is found within high-density lipoproteins (HDL), and alterations in its processing may likely lead to changes in lipid metabolism. This abnormal processing may reflect dysfunction of the exocrine process of the pancreas. However, the potential to use apoA2 as a biomarker exists regardless of the causative pathway. Hypo-processing of these lipoproteins is associated with increased dysplasia and malignant potential, likely to further alteration of lipid metabolism via its effect on HDL. When used as a biomarker to differentiate between low- and high-grade dysplasia, apoA2 performed better than CA19-9 [43].

As discussed above, *GNAS* alteration likely plays a key role in IPMN progression through various signaling pathways that regulate different cellular processes, including alteration of lipid metabolism. Mutant GNAS signaling promoted fatty acid oxidation and increased TAG and acetyl CoA levels [44]. Most cell membrane lipids are derived from acetyl CoA, and having a constant supply would support new cell production. *GNAS* mutations may very well be responsible for the overall increase in lipidomic pathways observed. However, further elucidations on the affected lipid pathways would help illuminate our understanding of the significance of the detected changes.

Other metabolic alterations include hormonal signaling. Leptin, a hormone that increases glucose uptake and fatty acid oxidation is increased by adipocytes. Leptin may also be altered in the progression of IPMN to PDAC and was found to be actually decreased in higher grades of dysplasia and invasive cancer as compared to low-grade dysplasia. Lower leptin levels were associated with cancer cachexia [45] and this likely explains the decreased levels seen in the progression of IPMN to PDAC [46].

Altered metabolites were proposed as key players in cellular signaling between cancerous and host cells. Metabolite cross feeding allows cancer cells to overcome limitations in nutrition and oxygen they may face within the tumor microenvironment. In PDAC, communication between fibroblasts and other cells in the microenvironment are thought to promote tumorigenesis [47]. Both cancer-associated fibroblasts (CAFs) and pancreatic stromal cells (PSCs) belonging to the host were implicated in pancreatic cancer as players in the complex cross feeding that occurs. Specifically, PSCs generate and secrete alanine which can then be utilized in the TCA cycle for nearby pancreatic cancer cells [48]. PSCs were also found to secrete lysophosphatidylcholines, a lipid that is used in cell membrane production as well as in synthesis of wound healing mediators [49]. PSCs are typically quiescent and store lipid droplets in their cytoplasm and, upon reception of activating signals from arising malignant cells, transform to CAFs and loose their lipid droplets [50]. A study tracing these lipids found significant accumulation of fatty acids once belonging to PSCs within PDAC cells, further supporting their role in supporting the metabolic needs of the cancer [49]. Via single cell RNA analysis, a significant increase in the presence of CAF was noted in high grade IPMN compared to low grade [51]. Across cancer types, CAFs were implicated in survival of cancerous cells in hypoxic conditions by helping to facilitate the switch from oxidative phosphorylation to aerobic glycolysis. This is in addition to observed dysregulation of amino acid and lipid metabolism, necessary for survival of the malignancy [52]. Mechanisms of cross talk between cancer cells and CAFs is beyond the scope of this paper, but as populations of CAFs grow with increased dysplasia of IPMN, they may contribute to the metabolic reprogramming that leads to malignancy prior to their role in maintaining cancer cell growth, progression, and immune evasion [52].

While the above changes in metabolites reflect the increased energy demand of increased cellular proliferation, changes in the expression of lipids and proteins can also enact change via their role in signaling. Another example of metabolite signaling is prostaglandin E2 (PGE_2_). PGE_2_ is a lipid that plays a key role in inflammation pathways. Derived from arachidonic acid released from the cell membrane and produced by the COX enzyme, PGE2 functions to increase local inflammation [53]. Both PGE2 and COX-2 were found to be elevated in various cancers including PDAC, and blocking the COX-2 pathway decreased inflammation and PDAC in mice [54]. When utilized as a potential biomarker, PGE2 concentrations in pancreatic cyst fluid were found to correlate with the grade of IPMNs. The PGE2 increase correlated with the level of dysplasia despite controlling for NSAID use, diabetes or pancreatitis, and cyst size [55]. This was validated in a larger study population; moreover, both cyst fluid PGE2 and IL-1β alone outperformed serum CA19-9 for predicting advanced neoplasia [56]. While not directly studied, one can hypothesize that the increase in lipidomic pathways would allow for increased production of cellular membrane lipids both for cellular growth and for release and use in signaling pathways such as COX-2 to form PGE2. The resulting inflammation from amplified signaling is an element of the altered microenvironment leading to dysplasia and cancer.

## 5. Inflammation

In addition to metabolic dysregulations, increased systemic inflammation was associated with multiple cancer malignancies. This is a plausible mechanism contributing to carcinogenesis. Cancer-triggered inflammation is thought to promote suppression of antitumor activity via modulation of the immune cell population as well as enabling malignant growth by providing growth factors and promoting angiogenesis and invasion. Inflammatory infiltrate is also thought to release reactive oxygen species which contribute to further mutations of nearby cells [57]. Several scores that integrate inflammation exist and were utilized in various malignancies as pre-intervention indicators of prognosis. Some were explored for its utility in PDAC, such as the Glasgow prognostic core, neutrophil-lymphocyte ratio, and prognostic nutrition index, although none are universally standard [58]. In addition to PGE2 and IL-1β discussed above, the utility of inflammatory markers as biomarkers for IPMN-associated malignancy was explored.

Neutrophil lymphocyte ratio (NLR), a marker of inflammation, is associated with worse outcomes in various malignancies, PDAC included. Neutrophilia and declining lymphocytes are thought to be indicative of increased inflammation and decreased immune surveillance. When elevated, patients had worse survival following resection [59]. Elevated NLR has utility in differentiating IPMN-associated invasive carcinoma and noninvasive disease. However, it is suboptimal to differentiate between grades of dysplasia, limiting the clinical value as a biomarker to stratify high risk lesions [60]. Sugimachi et al. suggested that in patients with a year or longer of preoperative surveillance time, higher NLR was significantly higher in IPMNs with advanced neoplasia as compared to those with low-grade dysplasia. This suggests that it still serves a purpose in guiding decisions regarding resection in patients who undergo surveillance for 12 or more months [61]. Predicting advanced neoplasia by combining NLR with another preoperative marker of inflammation, platelet to lymphocyte ratio (PLR), improves the prediction of advanced neoplasia [62].

Another proposed ratio indicative of systemic inflammation is the CRP to albumin ratio (CAR). It was investigated in patients with PDAC and found to be predictive of poor outcomes following resection, similar to NLR. When applied to IPMNs, low CAR predicted better survival. CAR also has potential utility as an independent predictor of advanced neoplasia in IPMN [63]. The CAR also gives some insight into the nutritional status of the patient. Elevated CRP indicates inflammation, both causative and necessary for malignant growth. Hypoalbuminemia is often observed in systemically and chronically ill patients and is associated with poor nutritional status and cachexia [64]. While metabolic pathways may be increased to support the growing malignancy, unfortunately, it does not support the nutritional status of the patient itself, only the growth of the malignancy within them.

If increased inflammation occurs with progression to cancer, biomarkers already associated with systemic inflammation may have utility in identifying high-risk IPMNs. Ferritin, a marker of acute and chronic inflammation, is also elevated in several malignancies. By modulating iron availability, ferritin may play a role in immunosuppressive modifications in myeloid and lymphocytes and contribute to overall immunosuppression in cancer patients [65]. It is also possible that ferritin exerts pro-oncogenic effects by promoting angiogenesis and encouraging cellular proliferation [66]. Serum ferritin was found to have similar predictivity as CA19-9 in assessing IPMN. As discussed above, when CA19-9 is elevated in cases of IMPN, while not diagnostic, it is another concerning data point when considering malignant risk [67]. 

## 6. Immune Modulation

Where there is inflammation, alterations in the immunological landscape also occur. While the specificities of this are beyond the scope of this paper, some of these alterations result in useful biomarkers for grades of dysplastic progression. T cell frequency and localization dramatically change. Cytotoxic T cells, activated Th cells, and dendritic cells infiltrate low-grade dysplastic IPMNs. Eventually, they are replaced by new immune cells with immune suppressive phenotypes, allowing progressive dysplasia and malignant growth without immune detection. CD8+ lymphocytes, significant players in antitumor immunity, when present, are associated with survival of the cancer patient and are sequestered off in peritumoral compartments, while more passive Tregs infiltrate the neoplasm microenvironment. In addition to Tregs, macrophages also are thought to begin to infiltrate low-grade dysplasia and numbers of both increase with progression. Higher levels of Tregs and macrophages within the local site of malignancy are a negative prognostic factor, as it correlates with poor immune detection of the malignant cells. Detection and quantification of these cell types at the site of the IPMN may help guide where in the pathway to malignancy a given IPMN is. In addition, Th cell subsets populations at the tumor site also change during progression. For example, Th1-polarized CD4+ cells are fewer than Th2-polarized CD4+ T cells with more advanced dysplasia. Similar to the inflammatory ratios discussed above, the ratio of these two Th1 cells show association with the grade of dysplasia [68]. When evaluated with multiplex immunofluorescence, immunohistochemistry, and high-resolution image analysis approaches, the promise of using alterations in the immune landscape as predictors of dysplasia was validated by Hernandez et al. Of note, the immune phenotype of low-grade IPMNs that had progressed at the time of surgical resection was similar to those with advanced neoplasia. In low-grade IPMNs at high risk for progression, the attenuated immune surveillance occurs earlier in the pathway of dysplasia. This offers a potential way to identify high-risk changes in immune surveillance and allow for risk stratification of the IPMN if micro-biopsied via endoscopic techniques although obtaining a biopsy with representative immune cell infiltrate provides its own challenges; it also provides for the potential of clinical interception at lower-grade dysplasia. If a vaccine or other modality is developed in the future to maintain or restore the local immune surveillance in IPMN to prevent the progression of IPMN to cancer, it would be potentially possible to intervene before the progression to advanced neoplasia necessitates surgical resection [69].

CAF, in addition to its role with altered metabolism and nutritional support, also is a player in immune evasion. As mentioned previously, populations of CAF were observed when RNA of 5403 cells of IPMN were analyzed by Bernard et al. Within the CAF population, two known different subtypes of CAF emerged. Inflammatory CAF (iCAF), characterized by lack of alpha smooth muscle actin (SMA) expression and secretion of IL6 and other inflammatory mediators such as IL-11 [70]. In contrast, myofibroblastic CAFs (myCAFs) mainly perform fibrogenesis [70]. IL-6 was found to be increased with PDAC, and was found to specifically suppress PPARalpha-regulated ketogenesis which may be correlated to tumor-induced cachexia [71]. IL-6 also functions to recruit an inflammatory immune population such as tumor-associated macrophages and contribute to a immunosuppressive microenvironment needed for evasion [72]. Populations of iCAF were identified only in invasive carcinoma, and were absent in non-invasive carcinoma, while myCAF were present even in low-grade dysplastic lesions and increased with progression to high-grade lesions, suggesting that CAFs’ role in inflammation and immune suppression occurs later in the IPMN to PDAC progression timeline [51]. More subtypes of CAF emerged with regard to PDAC, including mixed fibrogenic and inflammatory types, types without fibrotic or inflammatory features, and antigen-presenting subtypes (apCAF) [73,74]. It is thought that apCAF may act as a decoy receptor to deactivate CD4+ T cells and further contribute to immune suppression [74]. There is room for further investigation into CAFs’ role in IPMN’s progression to PDAC, as it may play a part in prepping the immune and inflammatory microenvironment in IPMN prior to its role in PDAC invasion and metastasis.

Prior to differentiating into CAF, PSC can become activated in comparison to their typical quiescent state in health. Activated PSC begin to express alpha smooth muscle actin, loose their lipid content, as discussed previously, and produce extracellular matrix proteins [75]. Included in these proteins are type I and III collagens, laminin, fibronectin, matrix metalloproteinases (MMPs), and tissue inhibitors of metalloproteinases [76]. This all contributes to desmoplasia, which includes the increased extracellular matrix protein, immune cells, activated PSCs and, eventually, CAF. Specific matricellular proteins produced by activated PSCs include periostin and galactin-1 were also observed in the stroma of PDAC, and their role in progression and invasion of PDAC were studied [77,78]. Stromal expression of alphaSMA was found to be increased in high grade dysplasia and IPMN with invasive carcinoma in a study considering PSCs role in IPMN progression to malignancy [79]. Additional markers of activated PSC including periostin and galactin-1 were reported to be higher in high-grade vs. low-grade dysplasia with regard to IPMN [79,80]. This suggests that activation of PSCs and their production of extracellular matrix proteins contributing stromal fibrosis is likely a contributing factor in the progression of IPMN to PDAC. The resulting desmoplasia characteristically surrounds PDAC cancer cells and it was proposed as a contributing factor to the severe and aggressive nature of pancreatic cancer. Supporting this thought, activated PSC were shown to promote pancreatic cell proliferation and invasion and decrease the effectiveness of radiation and chemotherapy compared to pancreatic cancer cells without the support of activated PSC [81]. As such, activated PSC not only have the ability to differentiate into CAF, but they may also play a role independently in IPMN malignant progression and the aggressiveness of the resulting cancer.

## 7. Clinical Applications

The majority of pancreatic cystic lesions are incidentally diagnosed initially on abdominal imaging. While definitive diagnosis is achieved via surgical resection or less invasive endoscopic ultrasound (EUS)-guided needle based laser confocal endomicroscopy (nCLE), cyst fluid molecular analysis by next-generation sequencing (NGS) analysis, or, in some instances, through-the-needle biopsy (TTNB), there is a need for further risk stratification of IPMNs [5,82,83]. Most experts follow the revised international consensus guidelines in the risk-stratification of IPMNs [5]. Specific criteria were suggested as worrisome and high-risk criteria. Worrisome features include imaging findings of cyst ≥ 3 cm, enhancing mural nodule < 5 mm, thickened enhanced cyst walls, an elevated serum level of CA 19-9, lymphadenopathy, abrupt change in main pancreatic duct caliber with distal pancreatic atrophy, pancreatic duct size of 5–9 mm, and a rapid rate of cyst growth > 5 mm/2 years. High-risk features include jaundice in a patient with a pancreatic cystic lesion of their pancreatic head, enhanced mural nodule ≥ 5 mm, pancreatic duct size ≥ 10 mm [5]. Throughout this review, several potential biomarkers obtained in the serum or analyzed from cyst fluid were mentioned and are summarized in Table 1 regarding their ability to predict level of dysplasia. Metabolic biomarkers, in particular products from the altered and increased lipidomic pathway, appear to hold great potential in prediction of dysplasia and may help to risk-stratify prior to resection [42]. 

In addition to obtainable biomarkers, imaging also remains important in initial diagnosis and monitoring of the progression in a non-invasive manner for patients who undergo surveillance of their lesions. Magnetic resonance imaging/magnetic resonance cholangiopancreatography (MRI/MRCP) and computed topography (CT) imaging both can offer non-invasive routes of surveillance. In a retrospective study comparing CT and MRI modalities of IPMN patients and their ability to identify the high risk features mentioned above according to the international consensus guidelines, CT and MRI performed similarly in their ability to identify malignant vs. non-malignant IPMN. However, MRI was able to identify enhancing mural nodules superiorly [84].

Liquid biopsy is another non-invasive tool that may aid in the diagnosis of high-grade IPMNs. The analysis of circulating cell-free DNA (cfDNA) isolated from blood samples was able to distinguish IPMN patients from controls. In addition, driver mutations including *KRAS* and *GNAS* were detected in the cfDNA of IPMN samples but not in benign lesions such as SCN [85]. This suggest that cfDNA analysis may help identify high-risk genetic alterations in IPMNs, thus avoiding the need for invasive biopsies. The analysis of extracellular vesicle (EV) derived from peripheral blood in patients with IPMN may be another potential tool for identifying invasive IPMN. In a recent study by Yang et al., the high level of Mucin 5AC (MUC5AC) in EV was highly predictive of invasive high grade IPMNs, while low levels were associated with low grade lesions [86].

In further efforts to limit unnecessary resections, advanced diagnostics via EUS procedures such as EUS-nCLE, NGS of cyst fluid, TTNB, and contrast enhanced harmonic endoscopic ultrasound (CH-EUS) were utilized. EUS-nCLE utilizes a laser scanning unit to also visualize the intracystic epithelium further and, in the case of IPMNs, reveal characteristic papillary projections. In a prospective study, EUS-nCLE was able to distinguish mucinous cyst vs. non-mucinous cyst with greater accuracy than measuring carcinoembryonic antigen (CEA) levels and cytology analysis [87]. Using computer aided diagnosis to analyze imaging taken during the endoscopy, EUS-nCLE has the potential to differentiate low-grade dysplasia from advanced neoplasia [82,83,88]. It also has the potential to differentiate between subtypes of IPMN pathologically [89]. Cyst fluid analysis by NGS is now more accurate in predicting advanced neoplasia than ever before [90]. One step further than sampling the cystic fluid is through the needle forceps biopsy (TTNB) which can be deployed endoscopically, although with a high adverse event rate [91]. As of now, lesions that meet high-risk criteria are sent for invasive surgical resection, but with the aforementioned endoscopic techniques being developed and validated, the possibility of limiting the frequency of invasive surgery may be possible.

## 8. Conclusions

IPMNs are often identified incidentally on imaging especially with advancing age. Accurate diagnosis and risk-stratification is critical to intervene prior to onset of invasive carcinoma and also to avoid unwarranted surgery. There is an ever increasing need to accurately grade the dysplasia and avoid surgical overtreatment of IPMNs; currently, nearly 50% of IPMNs resected have only low-grade dysplasia [92]. Routine imaging characteristics, EUS morphology, and cyst fluid analytics (CEA and cytology) have less than 50% accuracy in optimized risk-stratification of IPMNs [93]. Novel technologies including EUS-nCLE and cyst fluid NGS are not widely adapted.

As discussed in this review, the molecular and metabolic profiles driving IPMN tumorigenesis provided insight into potential biomarkers as well as novel therapeutic strategies. Molecular analysis of cystic fluid and tissue identified high risk mutations and novel drivers of IPMN tumorigenesis. Genomic alterations, in particular *KRAS* and *GNAS*, play an important role in mediating IPMN progression to PDAC. In addition to activating various signaling pathways that promote cell growth and survival, *KRAS* and *GNAS* mutation also results in downstream metabolic effects including lipid metabolism and glucose metabolism. Changes in the metabolic needs and altered pathways are both necessary for cancerous growth and may be measurable along the way to malignancy in the case of IPMN. Metabolic changes resulting in lipidomic pathway enhancement, in particular, show the promise for both a serum and cyst fluid biomarker. Not only may the altered metabolism be a key player in the progression of malignancy and perhaps allow for risk stratification of patients via utilization of metabolites as biomarkers, cross feeding and cross talk between cancerous cells and noncancerous cells likely lead to activation of PSC and eventual differentiation into CAF. These are critical for the desmoplasia, inflammation, and immune modulation and evasion that PDAC displays. Alterations within the microenvironment of IPMNs leading to increased inflammation and immune modulation also may allow for measurable differences in cell populations and inflammatory signals. Those reviewed had varying utility alone and may have increased utility if used in conjunction with other risk stratification tools outlined in the international consensus guidelines. Changes in the immune population are becoming better understood and offer a place of possible intervention to restore or keep the immune surveillance of the patient active, resulting in inability of PDAC to thrive. Continued characterization of precursor cystic lesions and the mechanisms underlying progression to PDAC may help identify new biomarkers which along with high-risk imaging features may more accurately guide IPMN management and possibly mitigate the need for unnecessary surgical intervention.

## Figures and Tables

**Figure 1 cancers-15-01722-f001:**
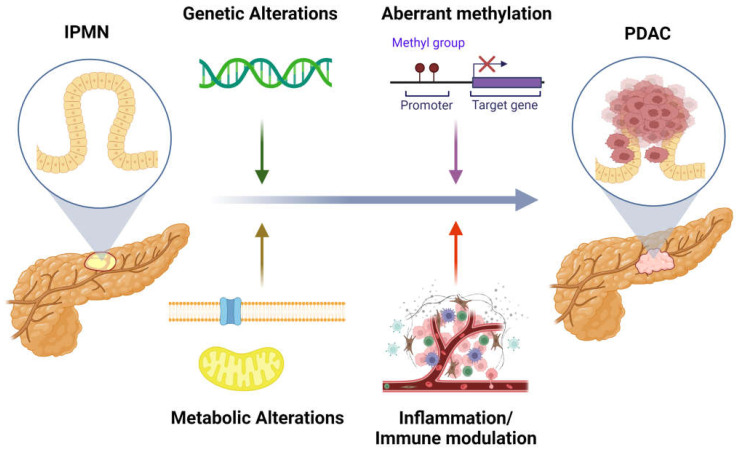
The impact of genetic, metabolic, inflammatory, and immunomodulatory mediators in progression of IPMN to PDAC. Created with BioRender.com.

**Table 1 cancers-15-01722-t001:** Utility of reviewed potential biomarkers in distinguishing grades of dysplasia.

Biomarkers	Biofluid	Predictor Value	Reference
CA 19-9	Serum	+	[28]
Lipidomic Pathway Products	Cyst fluid	+++	[42]
Altered TAGs	Serum	+	[42]
Altered TAGs	Cyst fluid	+++	[42]
Apolipoprotein A2	Serum	++	[43]
NLR	Serum	+	[60]
NLR with PLR	Serum	++	[61]
CAR	Serum	+	[63]
PGE2	Cyst fluid	++	[55]
IL-1β	Cyst fluid	++	[56]
Ferritin	Serum	+	[67]

Utility in distinguishing grades of dysplasia indicated with + weak predictor, ++ moderate predictor and +++ strong predictor.

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
