# Peer review of "The Role of Genetic, Metabolic, Inflammatory, and Immunologic Mediators in the Progression of Intraductal Papillary Mucinous Neoplasms to Pancreatic Adenocarcinoma"

_cancers, 2023, doi:10.3390/cancers15061722_

Round 1
Reviewer 1 Report
The review by Shockley et al provides a summary of the changes in the IPMNs that are important for understanding of the progression towards PDAC and of interest in improving early diagnosis. Unfortunately, it has some important gaps and needs to be improved to provide a comprehensive picture of the field.
First, the role of the activated stelate cells, or fibroblasts (some of which are antigen presenting) is completely lacking from the review. There are several reports that address this cell type and its role in PDAC progression from the precursors, including the sequencing data form the Ken Olive group.
Furthermore, apart from mutations, methylation changes have impact on the progression, and these have to be mentioned.
In terms of metabolism, KRAS mutation as a metabolic driver was not covered sufficiently, as well as the exchange of metabolites between the cancer-associated fibroblasts and cancer cells, for which there is plethora of literature (
.In the part about the metabolic alterations, there is a lack of discussion whether these changes can be tracked and monitored and whether they can be determined in serum or have to be inferred from neoplasms.
Another emerging field in early diagnosis and tracking of PDAC are ctDNA and egsosomes, and summary of findings on these features in the IPMN needs to be included.
In the clinical applications, invasive vs non invasive systems of monitoring of the progression also have to be highlighted.
Finally, the conclusion section has to be stronger in generalizing all of the parts of the review and the summary figure that models the progression from low to high grade IPMNs and the markers that are characteristic for each step based on the whole review would be of interest for the reader to better comprehend the conclusions.
Reviewer 2 Report
This review is well written and concise. The main topics addressed in this paper were covered in an comprehensive manner.
I have just one suggestion to make, in my opinion, this review more appealing and usable for a broader audience.
IPMN are rare tumors, or at least rarely detectable, and somehow fail to attract much attention, even in the researchers that work in pancreatic cancer. Therefore, I think it is necessary to add an intial paragraph to clearly explain the histological ( gastric, intestinal, pancreatobiliary, low-grade dysplasia vs high-grade dysplasia) and clinical ( low-risk vs high-risk, main duct vs branch duct) features of IPMN.
I saw that sometimes the genetic or metabolic features described in the review were associated to some of this entities and I fear that without an introductory paragraph few people can actually understand much. It would actually be interesting to delineate better how the described features also corellate with the malignancy of IPMN
I will suggest to take inspiration from this similar review PMID:32887490, that however treated different topics.
Good luck and thank you for this nice review
Round 2
Reviewer 1 Report
Authors have addressed all of my raised concerns and significantly improved the manuscript.